# "Acute kidney injury in critically ill patients with COVID–19: The AKICOV multicenter study in Catalonia"

**Arsenio De La Vega Sánchez**[1,2☯], **Ana Navas Pérez**[3☯], **Marcos Pérez-Carrasco**[1,2]*, **María Torrens Sonet**[4], **Yolanda Diaz Buendia**[5], **Patricia Ortiz Ballujera**[6], **Miguel Rodríguez López**[7], **Joan Sabater Riera**[8], **Aitor Olmo-Isasmendi**[9], **Ester Vendrell Torra**[10], **María Álvarez García-Pumarino**[11], **Mercedes Ibarz Villamayor**[12], **Rosa María Catalán Ibars**[13], **Iban Oliva Zelaya**[14], **Javier Pardos Chica**[15], **Conxita Rovira Anglès**[16], **Teresa M. Tomasa-Irriguible**[17], **Anna Baró Serra**[18], **Edward J. Casanova**[19], **Francisco J. González De Molina**[20☯], **on behalf of The AKICOV Group**[¶]

1 Intensive Care Department, Vall d'Hebron Hospital Universitari, Barcelona, Spain, 2 Shock, Organ Dysfunction and Resuscitation Research Group, Universitat Autònoma de Barcelona, Vall d'Hebron Research Institute (VHIR), Barcelona, Spain, 3 Critical Care Center, Parc Tauli Hospital Universitari, Institut d'Investigació i Innovació Parc Taulí I3PT, Universitat Autònoma de Barcelona, Sabadell, Barcelona, Spain, 4 Intensive Care Department, Hospital de la Santa Creu i Sant Pau, Barcelona, Spain, 5 Critical Care Department, Hospital del Mar, Barcelona, Spain, 6 Intensive Care Department, Hospital Universitari Doctor Josep Trueta de Girona, Girona, Spain, 7 Critical Care Department, Hospital del Baix Llobregat Moises Broggi Sant Joan Despí, Barcelona, Spain, 8 Intensive Care Department, Bellvitge University Hospital, L'Hospitalet de Llobregat, Barcelona, Spain, 9 Critical Care Department Hospital Universitari General de Catalunya Sant Cugat del Vallès, Sant Cugat del Vallès, Barcelona, Spain, 10 Intensive Care Department, Hospital General de Granollers, Granollers, Barcelona, Spain, 11 Consorci Sanitari de Terrassa Barcelona, Terrassa, Barcelona, Spain, 12 Critical Care Department, Hospital Universitari Sagrat Cor, Barcelona, Spain, 13 Critical Care Department, Hospital Universitari de Vic, Vic, Barcelona, Spain, 14 Hospital Universitari Joan XXIII de Tarragona, Tarragona, Spain, 15 Servicio Medicina Intensiva, Hospital Universitario Arnau de Vilanova de Lleida, Lleida, Spain, 16 Hospital Universitari Sant Joan de Reus, Reus, Tarragona, Spain, 17 Germans Trias i Pujol Hospital Intensive Care Department Badalona, Barcelona, Spain, 18 Critical Care Department Hospital Santa Caterina de Salt, Girona, Spain, 19 Critical Care Department Hospital Asepeyo Sant Cugat del Vallès, Sant Cugat del Vallès, Barcelona, Spain, 20 Intensive care Department, Mútua Terrassa University Hospital, Terrassa, Barcelona, Spain

☯ These authors contributed equally to this work.
¶ Membership of the AKICOV group is provided in the Acknowledgments.
* marcos.perez@vallhebron.cat

**Data Availability Statement:** All relevant data are within the manuscript.

**Funding:** The authors received no specific funding for this work.

## Abstract

This study describes the incidence, evolution and prognosis of acute kidney injury (AKI) in critical COVID-19 during the first pandemic wave. We performed a prospective, observational, multicenter study of confirmed COVID-19 patients admitted to 19 intensive care units (ICUs) in Catalonia (Spain). Data regarding demographics, comorbidities, drug and medical treatment, physiological and laboratory results, AKI development, need for renal replacement therapy (RRT) and clinical outcomes were collected. Descriptive statistics and logistic regression analysis for AKI development and mortality were used. A total of 1,642 patients were enrolled (mean age 63 (15.95) years, 67.5% male). Mechanical ventilation (MV) was required for 80.8% and 64.4% of these patients, who were in prone position, while 67.7% received vasopressors. AKI at ICU admission was 28.4% and increased to 40.1% during ICU stay. A total of 172 (10.9%) patients required RRT, which represents 27.8% of the

**Competing interests:** The authors have declared that no competing interests exist.

**Abbreviations:** ACE2, angiotensin-converting enzyme 2; AKI, Acute kidney injury; APACHE, Acute physiology and chronic health disease classification system; APPT, Activated partial thromboplastin time; ARDS, Acute respiratory distress syndrome; BMI, Body mass index; CKD, chronic kidney disease; COPD, chronic obstructive pulmonary disease; COVID-19, Coronavirus disease 2019; CRBSIs, Catheter-related bloodstream infections; eGFR, estimated glomerular filtration fraction; ICU, Intensive care unit; KDIGO, Kidney disease: Improving global outcomes; MDRD, Modification of diet in renal disease; MV, Mechanical ventilation; PT, Prothrombin time; RRT, Renal replacement Therapy; RT-PCR, Reverse transcription polymerase chain reaction; SARS-CoV-2, Severe acute respiratory syndrome coronavirus 2; SOFA, Sequential organ failure assessment; UTI, Urinary tract infections; VAP, Ventilator associated pneumonia; WHO, World health organization.

patients who developed AKI. AKI was more frequent in severe acute respiratory distress syndrome (ARDS) ARDS patients (68% vs 53.6%, p<0.001) and in MV patients (91.9% vs 77.7%, p<0.001), who required the prone position more frequently (74.8 vs 61%, p<0.001) and developed more infections. ICU and hospital mortality were increased in AKI patients (48.2% vs 17.7% and 51.1% vs 19%, p <0.001) respectively). AKI was an independent factor associated with mortality (IC 1.587–3.190). Mortality was higher in AKI patients who required RRT (55.8% vs 48.2%, p <0.04). **Conclusions** There is a high incidence of AKI in critically ill patients with COVID-19 disease and it is associated with higher mortality, increased organ failure, nosocomial infections and prolonged ICU stay.

## Introduction

Novel coronavirus infection is a contagious disease caused by severe acute respiratory syndrome coronavirus 2 (SARS-CoV-2). On 30 January 2020, the World Health Organization (WHO) declared the outbreak to be the sixth Public Health Emergency of International Concern. On 11 February 2020, the WHO named the disorder coronavirus disease 2019 (COVID-19).

Although it manifests mainly as an acute respiratory disease, Covid-19 can affect multiple organs, including the kidneys and the heart, the digestive tract, the blood, and the nervous system [1]. The first data from China suggested that fewer than 1% of patients with COVID-19 pneumonia had acute kidney Injury (AKI), however several subsequent studies have shown a considerable increase in renal impairment [2–4], and have linked this damage to an increase in in-hospital mortality [5]. These data have been supported by the presence of kidney damage in the autopsies of patients who died from the disease [6].

Several mechanisms related to kidney involvement in COVID-19 have been identified, including the tropism of the virus by the angiotensin-converting enzyme 2 (ACE2) receptor in kidney cells, damage caused by the storm of cytokines, which target different organs, and common causes of acute kidney damage in patients, such as drug toxicity, prerenal factors and other causes of AKI [7]. Therefore, kidney involvement in COVID-19 is multifactorial and can have a broad clinical spectrum, and mild kidney injury may go unnoticed.

Due to the pulmonary complications produced by SARS-CoV-2, many patients required admission to intensive care units (ICU), with an increase in the already high mortality rate [8, 9]. Furthermore, AKI is a common condition in critically ill patients and has been associated with even higher morbidity and mortality rates [10].

The aim of this study is to define the burden of kidney injury and its early prognosis in critical COVID-19 pneumonia admitted to 19 ICUs in Catalonia, Spain.

## Materials and methods

### Study population

In the present study, 1,642 patients admitted in ICUs who were diagnosed with COVID-19 from March 1 to May 15 were recruited from 19 hospitals in Catalonia. This study was approved by the ethics committee of the Hospital de Sabadell (Ref 2020/621). Verbal consent for participation was requested and recorded in the patient's medical history at each center.

## Inclusion criteria

Patients included were at least 18 years old, had laboratory-confirmed COVID-19 and had been admitted to a critical care area at one of the 19 hospitals in Catalonia. A confirmed case of COVID-19 was defined as a positive RT-PCR assay of a specimen collected via nasopharyngeal or oropharyngeal swab.

## Exclusion criteria

Patients were excluded if they were under 18 years old or had a non-COVID-19 related illness.

## Definitions and measurements

AKI was defined according to the Kidney Disease: Improving Global Outcomes (KDIGO) [11] criteria as follows. Stage I—An increase in serum creatinine of 0.3mg/dL within 48 hours or a 1.5 to 1.9 times increase in serum creatinine from baseline within seven days or an urine output <0.5ml/kg/h for six to twelve hours; Stage II–A 2 to 2.9 times increase in serum creatinine within seven days or urine output <0.5ml/k/h for more than 12 hours; Stage III–A threefold or more increase in serum creatinine within seven days, an increase of creatinine >4mg/dL, the initiation of renal replacement therapy (RRT) or urine output <0.3ml/k/h for more than 24 hours or anuria for more than 12 hours. Baseline creatinine was recorded from ambulatory data records.

ARDS was defined according to the Berlin definition [12], acute respiratory failure within one week of a known clinical insult or new or worsening respiratory symptoms, chest imagining with bilateral opacities- not fully explained by effusion, lobar/lung collapse, or nodules, respiratory failure not fully explained by cardiac failure or fluid overload and oxygenation criteria regarding $Pa\,O_2/FiO_2$: mild $200 < Pa\,O_2/FiO_2 < 300$ mmHg with PEEP or CPAP $\geq 5$ cmH$_2$O, moderate $100$ mmHg $< Pa\,O_2/FiO_2 < 200$ mmHg with PEEP $\geq 5$ cmH$_2$O, and severe $Pa\,O_2/FiO_2 < 100$ mmHg with PEEP $\geq 5$ cmH$_2$O.

Shock was defined as hypotension requiring vasopressors to maintain mean pressure higher than 65 mmHg, and have an elevated serum lactate (more than 2 mm/l) [13].

Hepatic failure was defined as increased serum bilirubin more than 1,9 mg/dL following criteria for liver failure used in the SOFA score [14].

Thrombocytopenia was defined less than 150.000 u/dL and coagulopathy an alteration of prothrombin time (PT) and activated partial thromboplastin time (APTT) in laboratory test made at ICU admission.

Neurological alteration at admission was defined as a mayor neurological alteration, excluding symptoms like headache, nausea, dizziness or vomiting. We consider neurological alteration altered mental state, encephalitis, encephalopathy or vascular disorder.

There was no unified protocol of treatment and all treatments, including RRT, were started by clinician decision.

## Data collection

The data were collected from on-line REDCap, to which a medical team added the information recorded at their hospitals. Data collected include demographic data, comorbidity including hypertension, chronic obstructive pulmonary disease (COPD) [15], asthma (GEMA 2012), diabetes mellitus [16], chronic kidney disease (CKD) any other disease diagnosed at admission, community treatment, APACHE II and SOFA scores, COVID-19 treatment received, AKI incidence and severity grade on the KDIGO/AKIN scale, the need for renal replacement

therapy, the need for mechanical ventilation, hemodynamic support, mortality and other events at admission.

## Statistical analyses

The descriptive statistical analyses performed include means and SDs for normally distributed continuous measures, medians and IQRs for skewed continuous measures, and frequency and proportions for categorical measures. We compared baseline patient characteristics between patients with and without AKI using the Chi-squared test or the Fisher exact test for categorical variables and the *Student's t-test* or the Mann-Whitney U test for continuous variables. A multivariate analysis was performed with logistic regression to identify the factors associated with AKI and mortality. All statistical tests were 2-sided, and a p-value <0.05 was considered statistically significant. Data analysis was conducted using StataCorp. 2019. *Stata Statistical Software*: *Release 16*. College Station, TX: StataCorp LLC.

## Results

### Demographics and characteristics of COVID-19 patients

The baseline characteristics of the COVID-19 patients with AKI admitted to the ICU compared with non-AKI patients are outlined in Table 1. Patients with AKI were significantly older than those in the non-AKI group (66 vs 60.7, p<0.001), a higher proportion were men (76.8% vs 61.3%, p<0.001) and they had higher severity score (APACHE II 15 vs 11, p<0.001 and SOFA 7 vs 4, p<0.001).

Regarding comorbidities, the most frequent pathology associated with COVID-19 was hypertension in 745 patients (57.9% vs 38.6%, p<0.001), followed by obesity (Body Mass Index (BMI) >30) (34.8%).

### ICU admission and organic dysfunction

At admission into the ICU, the diagnostic was viral pneumonia in 1,553 cases (94.6%). AKI (467 cases, corresponding to 28.4%) was the most frequent organic dysfunction in the group of COVID-19 patients, followed by shock in 447 cases (27.2%). All the organic dysfunctions at admission were significantly related with AKI (Table 2).

### COVID-19 treatment and coinfection at admission

Bacterial coinfection at admission to the ICU was not common in COVID-19 patients, with only 84 cases (5.3%), and with Streptococcus pneumoniae as the most frequent coinfection. During admission to the ICU, almost all patients received antibiotics 1439 (97.1%), with Azithromycin 1194 (72.7%) and Ceftriaxone 1095 (66.7%) being the most common. Piperacillin-Tazobactam and Meropenem were used in 469 (28.6%) and 375 (22.8%) cases respectively.

During the COVID-19 pandemic, different treatments were proposed (Table 3): hydroxychloroquine was used in 1,443 cases (90.8%), followed by Lopinavir/ritonavir in 949 cases (60.8%), other treatments being Tocilizumab in 527 cases (35.4%) and Remdesivir in 29 cases (2%). Treatment with corticosteroids was significantly more frequently used in AKI group (52.2% vs 47.4%, p = 0.048), especially as a treatment for shock in AKI group (7.1% vs 1%, p <0.001). There were no differences in other treatments between the AKI group and the non-AKI group, though interferon beta was more frequently used in the AKI group (169 cases, 29.5%) than in the non-AKI group (168 cases, 19.5%).

The use of corticosteroids during the COVID-19 pandemic was limited to other complications and not directly related to COVID-19 treatment. In this study, ARDS was the main

**Table 1. Patient characteristics at ICU admission.**

| General Data | Global | AKI | No AKI | p |
|---|---|---|---|---|
| | (n = 1642) | (n = 618, 40.1%) | (n = 923, 59.9%) | |
| Age median, year (IQR) | 63.0 (54.0,69.9) | 66.0 (58.8,71.7) | 60.7 (51.8,68.2) | <0.001 |
| Men n (%) | 1040 (67.5) | 474 (76.8) | 566 (16.3) | <0.001 |
| APACHE II median (IQR) | 12.0 (9.0,17.0) | 15.0 (11.0,20.0) | 11.0 (8.0,15.0) | <0.001 |
| SOFA score median (IQR) | 5.0 (3.0,7.0) | 7.0 (4.0,8.0) | 4.0 (3.0,6.0) | <0.001 |
| Comorbidities | | | | |
| Hypertension n (%) | 745 (45.5) | 358 (57.9) | 356 (38.6) | <0.001 |
| Asthma n (%) | 77 (4.7) | 20 (3.2) | 54 (5.9) | 0.0019 |
| COPD n (%) | 130 (7.9) | 71 (11.5) | 55 (6) | <0.001 |
| Heart failure n (%) | 16 (1) | 9 (1.5) | 6 (0.7) | 0.114 |
| Chronic kidney disease n (%) | 100 (6.1) | 76 (12.3) | 22 (2.4) | <0.001 |
| Chronic liver disease n (%) | 52 (3.2) | 26 (4.2) | 18 (2) | 0.009 |
| Pregnancy n (%) | 4 (0.2) | 1 (0.2) | 2 (0.2) | 1.00 |
| BMI median, (kg/m$^2$) (IQR) | 28.4 (25.7,32.0) | 28.7 (26.0,32.7) | 28.2 (25.6,31.6) | |
| Obesity BMI >30 n (%) | 517 (31.5) | 223 (42.5) | 278 (36.2) | 0.024 |
| Obesity BMI >40 n (%) | 54 (3.3) | 26 (5) | 28 (3.7) | 0.251 |
| Diabetes mellitus n (%) | 372 (22.7) | 190 (30.7) | 175 (19) | <0.001 |
| HIV infection | 2 (0.1) | 2 (0.3) | 0 (0) | 0.161 |
| Chronic coronary syndrome n (%) | 76 (4.6) | 41 (6.6) | 34 (3.7) | 0.008 |
| Autoimmune disease n (%) | 53 (3.2) | 23 (3.7) | 28 (3) | 0.459 |
| Community Treatment | | | | |
| ACEi n (%) | 345 (21) | 176 (28.5) | 158 (17.1) | <0.001 |
| ARBs n (%) | 200 (12.2) | 94 (15.2) | 96 (10.4) | 0.005 |
| β-blockers n (%) | 149 (9.1) | 83 (13.4) | 62 (6.7) | <0.001 |
| Diuretics n (%) | 178 (10.8) | 94 (15.2) | 78 (8.5) | <0.001 |
| NSAID´s n (%) | 79 (4.8) | 38 (6.1) | 41 (4.4) | 0.136 |
| Statins n (%) | 377 (23) | 181 (29.3) | 184 (19.9) | <0.001 |
| Anticoagulant therapy n (%) | 60 (3.7) | 32 (5.2) | 26 (2.8) | 0.017 |

IQR- Interquartile range. BMI- Body Mass Index

AKI- Acute kidney injury

COPD- Chronic obstructive pulmonary disease

HIV- Human immunodeficiency virus

ACEi- Angiotensin-converting enzyme inhibitors

ARBs- Angiotensin II receptor blockers

NSAIs- Nonsteroidal anti-inflammatory drugs.

reason for their use and they were used in 645 (39.3%) of the patients with no difference between groups. Other indications were shock and asthma or bronchial hyperreactivity.

Significantly more patients with AKI required invasive mechanical ventilation, namely 568 patients (91.9%) vs 717 (77.7%), of whom 462 (74.8%) needed to be placed in the prone position in the AKI group, which is significantly more than in the non-AKI group. Ventilation with high flow nasal cannula was more common in the non-AKI group.

Hemodynamic support was needed during ICU admission. Noradrenaline was the most frequently used treatment for hypotension and its use was significantly different between the two groups (AKI group: 518 (83.8%) vs 560 (60.7%) in the non-AKI group, p<0.001). Likewise, the use of Dobutamine was higher in the AKI group than in the non-AKI group. Other supports, such as Adrenaline, Dopamine and Levosimendan, were used in selected patients.

**Table 2. ICU admission and organic dysfunction.**

| | Global | AKI | No AKI | p |
|---|---|---|---|---|
| | (n = 1642) | (n = 618, 40.1%) | (n = 923, 59.9%) | |
| **Diagnosis at admission** | | | | |
| Viral pneumonia n (%) | 1553 (94.6) | 600 (97.1) | 899 (97.4) | 0.712 |
| COPD exacerbation n (%) | 16 (1) | 8 (1.3) | 8 (0.9) | 0.417 |
| Acute asthma n (%) | 17 (1) | 5 (0.8) | 12 (1.3) | 0.366 |
| Heart failure n (%) | 23 (1.4) | 14 (2.3) | 9 (1) | 0.041 |
| Other n (%) | 47 (2.9) | 134 (21.7) | 103 (11.2) | 0.362 |
| **Severity of the ARDS** | | | | <0.001 |
| Mild n (%) | 127 (7.9) | 28 (4.6) | 92 (10.1) | |
| Moderate n (%) | 514 (32.7) | 168 (27.5) | 330 (36.3) | |
| Severe n (%) | 935 (59.4) | 416 (68) | 488 (53.6) | |
| **Organic dysfunction at admission** | | | | |
| Shock n (%) | 447 (27.2) | 276 (44.7) | 155 (16.8) | <0.001 |
| AKI n (%) | 467 (28.4) | | | |
| Hepatic failure n (%) | 291 (17.7) | 158 (25.6) | 131 (14.2) | <0.001 |
| Coagulopathy n (%) | 245 (14.9) | 134 (21.7) | 103 (11.2) | <0.001 |
| Neurological alteration n (%) | 189 (11.5) | 102 (16.5) | 83 (9) | <0.001 |
| Thrombocytopenia n (%) | 169 (19.3) | 100 (16.2) | 68 (7.4) | <0.001 |

COPD- Chronic obstructive pulmonary disease

ARDS- Acute respiratory distress syndrome

AKI- Acute kidney injury.

## Acute kidney injury

Acute kidney injury was the most common complication at admission to the ICU (Table 4). Baseline creatinine was higher in the AKI group (0.9 vs 0.76, p<0.001), creatinine at hospital and ICU admission was significantly higher in the AKI group. The use of nephrotoxic drugs was higher in the AKI group (50.5% vs 41.4%, p<0.001). AKI evolution during admission was transitory AKI (<72h) in 188 (31.1%) patients, between 72 hours and seven days in 209 (34.5%) and more than seven days in 208 (34.4%) patients. Creatinine levels at hospital discharge were higher in the AKI group (0.96 vs 0.62, p<0.001). A total of 172 (10.9%) patients required renal replacement therapy, which represents 27.8% of the patients who developed AKI, of whom 151 (91.5%) had a KDIGO stage III AKI. Anuric state and hypervolemia were the most frequent indications for RRT.

## ICU complications and outcomes

During admission to the ICU, multiple complications were detected in COVID-19 patients. In our study, we focused on infectious and thrombotic events (Table 5). Ventilator-associated pneumonia (VAP) was the most common complication in our study with 477 cases (30.5%). The AKI group had significantly more cases of VAP than the non-AKI group (37.7% vs 25.6%, p<0.001). Catheter-related bloodstream infections (CRBSIs) occurred in 280 patients (18%) and urinary tract infections (UTIs) in 229 (14.8%). Thrombotic events, such as acute pulmonary embolism and deep vein thrombosis, were detected in 236 (15.3%) patients, with differences between groups.

ICU and hospital length of stay (LOS) were also analyzed, and there were significant differences between the AKI group and the non-AKI; hospital LOS (28 vs 24 days, p 0.002) and ICU LOS (18 days vs 11 days, p <0.001).

**Table 3. Treatments and support.**

| | Global | AKI | No AKI | p |
|---|---|---|---|---|
| | **(n = 1642)** | **(n = 618, 40.1%)** | **(n = 923, 59.9%)** | |
| **COVID-19 treatment** | | | | |
| Hydroxychloroquine/ Chloroquine. n (%) | 1443 (90.8) | 553(89.8) | 840 (91.2) | 0.345 |
| Lopinavir/Ritonavir. n (%) | 949 (60.8) | 358 (60.1) | 552 (60.4) | 0.899 |
| Remdesivir. n (%) | 29 (2.0) | 11 (2.0) | 18 (2.1) | 0.925 |
| Interferon b. n (%) | 352 (23.5) | 169 (29.5) | 168 (19.1) | <0.001 |
| Tocilizumab. n (%) | 527 (35.4) | 182 (32.6) | 327 (37.2) | 0.073 |
| **Corticosteroids** | 756 (49.2) | 310 (52.6) | 425 (47.4) | 0.048 |
| for shock n (%) | 53 (3.2) | 44 (7.1) | 9 (1.0) | <0.001 |
| for ARDS n (%) | 645 (39.3) | 248 (40.1) | 378 (41.0) | 0.747 |
| for asthma/bronchial hyperreactivity n (%) | 67 (4.1 | 27 (4.4) | 39 (4.2) | 0.891 |
| **Hemodynamic support** | | | | |
| Noradrenaline n (%) | 1112 (67.7) | 518 (83.8) | 560 (60.7) | <0.001 |
| Dobutamine n (%) | 27 (1.6) | 20 (3.2) | 6 (0.7) | <0.001 |
| Dopamine n (%) | 7 (0.4) | 4 (0.6) | 3 (0.3) | 0.448 |
| Adrenaline n (%) | 11 (0.7) | 8 (1.3) | 3 (0.3) | 0.033 |
| Levosimendan n (%) | 2 (0.1) | 1 (0.2) | 1 (0.1) | 1.000 |
| **Respiratory Support** | | | | |
| Oxygen n (%) | 924 (56.33) | 349 (56.5) | 547 (59.3) | 0.276 |
| High-flow nasal cannula n (%) | 651 (39.6) | 214 (34.6) | 412 (59.3) | <0.001 |
| Non-invasive ventilation n (%) | 307 (18.7) | 125 (20.2) | 172 (18.6) | 0.438 |
| Mechanical ventilation n (%) | 1326 (80.8) | 568 (91.9) | 717 (77.7) | <0.001 |
| Prone position n (%) | 1058 (64.4) | 462 (74.8) | 563 (61) | <0.001 |
| ECCO2R n (%) | 7 (0.4) | 5 (0.8) | 2 (0.2) | 0.124 |
| ECMO n (%) | 42 (2.9) | 19 (3.1) | 23 (2.5) | 0.491 |

ARDS- Acute respiratory distress syndrome

ICU- Intensive care unit

ECCO2R- extracorporeal membrane carbon dioxide removal

ECMO- Extracorporeal membrane oxygenation

AKI- Acute kidney injury.

COVID-19 related mortality ICU was significantly higher in the AKI group than in the non-AKI group (48.2% vs 17.7%, p<0.001). Moreover, overall mortality was significantly higher in the AKI group (51.1% vs 172 19%, p<0.001) and in patients who required RRT (55.8%).

## Discussion

This is the first multicenter study of 19 ICUs in a specific region during the first wave of the pandemic that presents the characteristics of COVID-19 critically ill patients who developed AKI, showing increased mortality and hospital stay. Catalonia was one of the most affected regions in Spain, with a total of 55,196 COVID-19 infections and a high volume of patients requiring ICU admission [17].

Initially, several studies suggested that AKI was not frequently seen in COVID-19 patients [18], being in accordance with the progress of the pandemic as one of the most major complications [19, 20]. Other complications showed major differences between studies: for example, Klok FA et al. [21] reported a 31% rate of thrombotic events, while others, such as Gupta A

**Table 4. Characteristics of acute kidney injury.**

| | Global | AKI | No AKI | p |
|---|---|---|---|---|
| | (n = 1642) | (n = 618, 40.1%) | (n = 923, 59.9%) | |
| Baseline creatinine | 0.80(0.27) | 0.90 (0.32) | 0.76 (0.29) | <0.001 |
| Creatinine at admission | 0.87 (0.43) | 1.10 (0.64) | 0.80 (0.27) | <0.001 |
| ICU admission creatinine | 0.80 (0.46) | 1.18 (0.71) | 0.70 (0.27) | <0.001 |
| Creatinine at discharge | 0.7 (0.47) | 0.96 (1.08) | 0.62 (0.28) | <0.001 |
| Use of nephrotoxics n (%) | 711 (43.4) | 312 (50.5) | 382 (41.4) | <0.001 |
| AKI evolution | | | | |
| Transitory <72h n (%) | | 188 (31.1) | | |
| >72h to < 7d n (%) | | 209 (34.5) | | |
| >7d n (%) | | 208 (34.4) | | |
| RRT n (%) | 172 (10.9) | 172 (27.8) | | |
| RRT indication | | | | |
| Hypervolemia n (%) | | 85 (49.4) | | |
| Pulmonary edema n (%) | | 3 (1.7) | | |
| Anuric n (%) | | 109 (63.4) | | |
| Hyperkalemia n (%) | | 56 (32.6) | | |
| Uremic syndrome n (%) | | 52 (30.2) | | |
| Sepsis n (%) | | 32 (18.6) | | |
| Severity of AKI at RRT start | | | | |
| KDIGO I n (%) | | 4 (2.4) | | |
| KDIGO II n (%) | | 10 (6.1) | | |
| KDIGO III n (%) | | 151 (91.5) | | |

AKI- Acute kidney injury

RRT- Renal replacement therapy

h-Hour. d-days.

et al. [20], reported a 10.3% rate of thrombotic events. Myocardial injury, with an elevation of cardiac biomarkers, occurred in 20–30% of hospitalized patients with COVID-19, with higher rates (55%) among those with pre-existing cardiovascular disease [22, 23]. Among AKI patients, the presence of another organic dysfunction such as liver failure, coagulopathy or

**Table 5. Complications and outcome.**

| | Global | AKI | No AKI | p |
|---|---|---|---|---|
| | (n = 1642) | (n = 618, 40.1%) | (n = 923, 59.9%) | |
| **Complications** | | | | |
| Ventilator-associated pneumonia n (%) | 477 (30.5) | 229 (37.7) | 231 (25.6) | <0.001 |
| Catheter-related bloodstream infections n (%) | 280 (18.0) | 126 (20.9) | 145 (16.0) | 0.015 |
| Urinary tract infections n (%) | 229 (14.8) | 112 (18.6) | 111 (12.4) | 0.001 |
| Thrombotic events n (%) | 236 (15.3) | 109 (18.1) | 118 (13.2) | 0.010 |
| **Outcome** | | | | |
| Length stay in ICU median (IQR) | 14.0 (6.0,27.0) | 18.0 (8.0,33.0) | 12.0 (5.0,24.0) | <0.001 |
| Length stay in Hospital median (IQR) | 26.5 (15.0,44.0) | 29.0 (16.0,52.0) | 25.0(15.0,39.0) | 0.002 |
| ICU mortality n (%) | 469 (29.7) | 296 (48.2) | 162 (17.7) | <0.001 |
| Global mortality n (%) | 493 (31.2) | 311 (51.1) | 172 (19) | <0.001 |

ICU–Intensive Care Unit

IQR- Inter Quartile Range.

neurological disorder was more frequent. Additionally, those patients were more likely to require mechanical ventilation and develop a severe ARDS. All these conditions coincide with higher severity scores (APACHE II and SOFA) at admission, similar to the patients in our study.

Although this study includes only critically ill patients admitted to ICU, AKI incidence is similar to that described by Hirsch JS et al. [24] who found an incidence of 36% among the 5,449 patients admitted to hospitals in New York from March 1 to April 5, 2020. Furthermore, a total of 1,395 (25.6%) were critically ill patients, most of them required mechanical ventilation and 1,060 patients (75%) developed AKI. Piñeiro et al. [25] analyzed 52 critically ill patients with COVID-19 and AKI admitted to ICU, our AKI incidence is higher (40.1% vs 21.4%), by highlighting more patients with BMI >30 (42.5% vs 21.2%) and younger (66y vs 71y), despite this differences RRT requirement and AKI mortality are similar (27.5% vs 28.5%; 51.1% vs 52%). Zheng X et al. [26] described that kidney injury of some kind occurred in 42% of the two cohorts that they analyzed, though only 6% of those patients developed a KDIGO criteria of AKI. Among critically ill patients, 30% developed AKI during admission and 9% had pre-hospital AKI. In another study analysing AKI incidence in COVID patients Chung M et al. [27] describe that AKI occurred in 1,406 (46%) patients overall, when referring to critically ill patients the incidence rises to 68% (admission plus new cases).

Finally, a multi-center study in Belgium [28] describes 85.1% of AKI incidence in critically ill patients. These large differences between studies may be explained by the important differences between the populations analyzed and the different definitions of AKI used in the studies. Hirsch JS et al. [24] used the KDIGO score but not urine output due to a lack of data, while an algorithm was used to calculate baseline creatinine because only 15% had records with baseline creatinine [17]. Chung M et al. [27] define AKI as at least an increase in peak serum creatinine of 0.3 mg/dL or 50% above baseline, while for patients without a baseline creatinine measurement, creatinine at admission was imputed based on a Modification of Diet in Renal Disease (MDRD) estimated glomerular filtration fraction (eGFR) of 75 ml/min/1.73m as per Kidney Disease: Improving Global Outcomes (KDIGO) AKI guidelines [11]. Higher incidence in the Belgium cohort could partially explain by the use of the urinary output criteria and the high proportion of obese patients in the AKI group (one-third) and the use of actual body weight for calculation of UO as mL/kg/h.

In our study, hypertension, COPD, BMI >30, CKD, and diabetes mellitus were the comorbidities associated with the AKI group, and these findings are consistent with those described in previous studies [4, 29, 30]. Obesity was the most frequent independent comorbidity in critically ill COVID patients who developed AKI.

Different treatments have been proposed during the pandemic. In our study, patients received Hydroxychloroquine, Beta interferon, Lopinavir/ritonavir, Tocilizumab and 29 patients were administered Remdesivir. None of these therapies has been related with a greater risk of AKI in previous studies and there was no difference in our study between the AKI group and the non-AKI group. Remdesivir and AKI were analyzed by Wang et al. but no relation was observed. Lopinavir/ritonavir has been related with acute interstitial nephritis in HIV patients but not with COVID-19 patients [31].

During the first wave of the COVID-19 pandemic, patients underwent significant changes in treatment protocols. However, our study found no differences between ICUs, as a unified protocol for regional assistance and a controlling center for resource distribution were in place. Antibiotics were commonly prescribed upon admission and were selected based on the local ecology of each medical center, but no significant differences were detected.

In our study, corticosteroid use was more frequent in the AKI group. The relationship between corticosteroid treatment and AKI in patients with COVID has not been directly

related. Cheng et al. [5] found that patients with AKI were more likely to receive glucocorticoids on admission, along the same lines as critically ill patients [19]. Moreover, their use has been related to a decrease in the requirement of RRT [16]. In our case, the administration of corticosteroids is related to the presence of shock and ARDS, more frequent in patients who developed AKI. Unfortunately, the relationship between corticosteroids and AKI in critically ill COVID 19 remains unresolved.

A small number of studies have focused on RRT, most of which are single center studies. Most of the data about RRT has linked its use to a higher mortality rate and more severe illness [32]. Focusing on COVID and RRT, Gupta S et al. [33] describe the results of 3,099 critically ill patients with COVID19 in the United States, describing how 20.7% of the patients developed AKI-RRT and had a mortality rate above 60%. In our study, the need for RRT in critically ill patients admitted to the ICU was only 10.9%, similar to Belgium cohort [28], this could be explained by a conservative approach for initiate RRT in COVID19 patients. Mortality rate in COVID patients with AKI-RRT is 55.8%, similar to other studies [32, 33].

AKI and mortality were widely analyzed by Lim MA et al. [34] in a systematic review and meta-analysis that concluded that AKI had a RR (RR: 13.38 [8.15, 21.95]), Chung M et al. [27] conclude that in-hospital mortality in patients with AKI was 41% overall and 52% in intensive care. In this study, mortality rate (51.1%) is similar to that described in those studies when referring to critically ill patients.

This study has some limitations. Firstly, it is a descriptive study focusing on a single geographical region. Therefore, it may be difficult to generalize these findings to other regions. Secondly, the diagnosis of AKI was mainly based on creatinine and oliguria variation, without taking into account variations in urine output. Thirdly, there was no unified protocol for the treatment of COVID-19 among hospitals, no for the prevention, management of AKI or use of RRT. Finally, the study describes what happened during the first pandemic wave in which there was no standardized treatment, so there could be differences with the results obtained in the following pandemic waves in which treatment was standardized.

## Conclusion

This study represents one of the largest series of critically ill COVID-19 with AKI in a specific region of Europe during the first pandemic wave, confirming their increased rate of mortality and ICU and hospital stay. Furthermore, it suggests that hypertension, COPD, BMI >30, CKD and diabetes mellitus are comorbidities associated with AKI in these patients; requiring RRT would increase their mortality.

Finally, studies involving multiple regions may be needed to understand the true incidence and involvement of AKI in critically ill patients with COVID-19.

## Ethics approval

This study was approved by the ethics committee of the Hospital de Sabadell (Ref 2020/621.)

## Acknowledgments

We would like to thank the members of the AKICOV group and the respective study collaborators at each intensive care unit: Carolina Maldonado and Abrahan Mera, Department of Critical Care, Vall d´Hebron University Hospital; Mireia Cerdà Martínez, Loreley Betancourt Castellanos, Gemma Gomà Fernández, Critical Care Center, Parc Tauli Hospital Universitari; María del Mar Fernández, Enrique Piacentini Intensive care Department, Mútua Terrassa University Hospital; Ana Segarra Martínez-Sahuquillo, Intensive Care Department. Hospital de la Santa Creu i Sant Pau; Anna Horta Puig, Intensive Care Department. Hospital

Universitari Doctor Josep Trueta de Girona; Susana Hernandez Marín, Beatriz Cancio Rodriguez and Arantxa Mas Serra, Critial care department. Hospital del Baix Llobregat Moises Broggi; Miguel Angel Gordillo Benítez, Intensive Care department. Bellvitge university hospital; Sandra Barbadillo Ansorregui, Critical Care department Hospital Universitari General de Catalunya Sant Cugat del Vallès; Cristina Pedrós Mas, Intensive Care Department. Hospital General de Granollers; Federico Esteban Reboll, Hospital Universitari Joan XXIII de Tarragona; Laura Raguer-Pardo, Germans Trias i Pujol Hospital Intensive Care Department.

## Author Contributions

**Conceptualization:** Marcos Pérez-Carrasco, Francisco J. González De Molina.

**Data curation:** Ana Navas Pérez, María Torrens Sonet, Yolanda Diaz Buendia, Patricia Ortiz Ballujera, Miguel Rodríguez López, Joan Sabater Riera, Aitor Olmo-Isasmendi, Ester Vendrell Torra, María Álvarez García-Pumarino, Mercedes Ibarz Villamayor, Rosa María Catalán Ibars, Iban Oliva Zelaya, Javier Pardos Chica, Conxita Rovira Anglès, Teresa M. Tomasa-Irriguible, Anna Baró Serra, Edward J. Casanova.

**Formal analysis:** Francisco J. González De Molina.

**Supervision:** Ana Navas Pérez, Marcos Pérez-Carrasco, Francisco J. González De Molina.

**Writing – original draft:** Arsenio De La Vega Sánchez.

**Writing – review & editing:** Ana Navas Pérez, Marcos Pérez-Carrasco, Francisco J. González De Molina.

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
