## [Decision Letter · Decision Letter 0]

24 Feb 2023

PONE-D-22-33365Acute kidney injury in critically ill patients with COVID–19: the AKICOV multicenter study in CataloniaPLOS ONE

Dear Dr. Pérez-Carrasco,

Thank you for submitting your manuscript to PLOS ONE. After careful consideration, we feel that it can be accepted after slight changes suggested by the Reviewer. Therefore, we invite you to submit a revised version of the manuscript that addresses the points raised during the review process.

We look forward to receiving your revised manuscript.

Kind regards,

Jordi Camps

Academic Editor

PLOS ONE

Journal Requirements:

Additional Editor Comments (if provided):

The article can be accepted after taking into account the comments of the reviewer

Reviewers' comments:

Reviewer's Responses to Questions

**Comments to the Author**

1. Is the manuscript technically sound, and do the data support the conclusions?

Reviewer #1: Yes

2. Has the statistical analysis been performed appropriately and rigorously? 

Reviewer #1: Yes

3. Have the authors made all data underlying the findings in their manuscript fully available?

Reviewer #1: Yes

4. Is the manuscript presented in an intelligible fashion and written in standard English?

Reviewer #1: Yes

5. Review Comments to the Author

Reviewer #1: A very interesting study.

Avoid abbreviations not explained in the abstract.

Also in the text when using the abbreviations explain them the first time they are used, see RRT.

The material and method need to be defined: ARDS, organic dysfunction,

Describing the usual treatments of the patients and those used during admission brings originality to the study, there are few that describe them.

Have differences been observed between the participating hospitals in terms of these treatments?

6. PLOS authors have the option to publish the peer review history of their article (what does this mean?). If published, this will include your full peer review and any attached files.

Reviewer #1: No

---

## [Author Response · Author response to Decision Letter 0]

24 Mar 2023

Dear Reviewers of PLOS One,

We would like to express our gratitude for taking the time to review our article titled "Acute kidney injury in critically ill patients with COVID-19: the AKICOV multicenter study in Catalonia." Your constructive feedback and suggestions were extremely helpful in improving the quality of the article.

We are writing to inform you of the changes we made based on your feedback:

1- “Avoid abbreviations not explained in the abstract” and “Also in the text when using the abbreviations explain them the first time they are used, see RRT”. We have explained all abbreviations used in the abstract and manuscript, as suggested in your comments.

2- “The material and method need to be defined: ARDS, organic dysfunction”. We included definitions for all the dysfunctions used in the study, as you recommended.

3- “Describing the usual treatments of the patients and those used during admission brings originality to the study, there are few that describe them”. We have expanded the discussion section to include a more detailed description of the usual treatments given to patients and those used during admission, as you noted that few studies have done so.

4- “Have differences been observed between the participating hospitals in terms of these treatments?”. We have explained in the discussion that no significant differences were detected between ICUs, likely due to a regional unified protocol being established.

Yours faithfully,

Marcos Pérez Carrasco Arsenio de la Vega Sánchez

---

## [Editor Report · Decision Letter 1]

27 Mar 2023

Insuficiencia renal aguda en pacientes críticos con COVID-19: el estudio multicéntrico AKICOV en Cataluña

PONE-D-22-33365R1

Dear Dr. Pérez-Carrasco,

We’re pleased to inform you that your manuscript has been judged scientifically suitable for publication and will be formally accepted for publication once it meets all outstanding technical requirements.

PD: The title has been written in Spanish in the web page, although it is in English in the manuscript. Please, correct.

Kind regards,

Jordi Camps

Academic Editor

PLOS ONE
---

## [Editor Report · Acceptance letter]

5 Apr 2023

PONE-D-22-33365R1 

“Acute kidney injury in critically ill patients with COVID–19: the AKICOV multicenter study in Catalonia” 

Dear Dr. Pérez-Carrasco:

I'm pleased to inform you that your manuscript has been deemed suitable for publication in PLOS ONE. Congratulations! Your manuscript is now with our production department. 

Kind regards, 

on behalf of

Dr. Jordi Camps 

Academic Editor

PLOS ONE